# The Cost-Effectiveness Analysis of Cervical Cancer Screening Using a Systematic Invitation System in Lithuania

**DOI:** 10.3390/ijerph16245035

**Published:** 2019-12-11

**Authors:** Justina Paulauskiene, Mindaugas Stelemekas, Rugile Ivanauskiene, Janina Petkeviciene

**Affiliations:** 1Department of Preventive Medicine, Faculty of Public Health, Medical Academy, Lithuanian University of Health Sciences, 47181 Kaunas, Lithuania; mindaugas.stelemekas@lsmuni.lt (M.S.); rugile.ivanauskiene@lsmuni.lt (R.I.);; 2Health Research Institute, Faculty of Public Health, Lithuanian University of Health Science, 47181 Kaunas, Lithuania

**Keywords:** cost-effectiveness, cervical cancer screening, invitation methods

## Abstract

In Lithuania, cytological screening of cervical cancer (CC) is largely opportunistic. Absence of standardized systematic invitation practice might be the reason for low participation rates. The study aimed to assess the cost-effectiveness of systematic invitation approach in CC screening programme from the perspective of a healthcare provider. A decision tree was used to compare an opportunistic invitation by a family doctor, a personal postal invitation letter with appointment time and place, and a personal postal invitation letter with appointment time and place with one reminder letter. Cost-effectiveness was defined as an incremental cost-effectiveness ratio (ICER) per one additionally screened woman and per one additional abnormal Pap smear test detected. The ICER of one personal postal invitation letter was €9.67 per one additionally screened woman and €55.21 per one additional abnormal Pap smear test detected in comparison with the current screening practice. The ICER of a personal invitation letter with an additional reminder letter compared to one invitation letter was €13.47 and €86.88 respectively. Conclusions: A personal invitation letter approach is more effective in increasing the participation rate in CC screening and the number of detected abnormal Pap smears; however, it incurs additional expenses compared with current invitation practice.

## 1. Introduction

Cervical cancer (CC) is the fourth most commonly occurring cancer with fatal outcome in women worldwide [1]. In 2018, the estimated age-standardized mortality rates ranged from about 2/100,000 in Western and Northern Europe to more than 19/100,000 in some African countries [1]. In Lithuania, the mortality rate from CC remains one of the highest among the Member States of the European Union (EU)—9.6/100,000 in 2018 [2]. CC is the second most common female cancer and the first leading cause of cancer deaths in Lithuanian women aged 15 to 44 years [3]. 

In 2003, the Council of the EU adopted recommendations on cancer screening, which rely on a population-based organized approach with appropriate quality assurance at all the levels [4,5]. Evidence suggests that a population-based cytological screening programme is an effective method to reduce the incidence and mortality rates of CC [6,7]. The success of CC screening programme depends on screening coverage [8]. CC screening programmes implemented in organized population-based settings ensure greater participation of the target population compared with opportunistic screening, which relies on the frequency of visits to a family doctor and doctors’ actions [9,10]. High rates of CC in Central and Eastern European countries, including Lithuania, demonstrate the lack of effective organized screening in these countries [8].

In 2004, Lithuania started the Nationwide Cervical Cancer Screening Programme, which is financed by the National Health Insurance Fund under the Ministry of Health of Lithuania [11]. This programme offers a free Pap smear test every 3 years to women aged 25–60 years. Primary healthcare centres (PHCC) are responsible for inviting women as well as taking a Pap smear test. Each centre has a right to choose their method of invitation for a Pap smear test: a verbal invitation during a family doctor’s visit or by phone, a written postal invitation or an invitation by SMS text message. Only a few PHCC send out postal personal invitations with personal screening information to target population women, which is a common method in organized (not opportunistic) national population screening programmes. A diversity of invitation methods as well as the opportunistic approach does not assure adequate participation rate and screening coverage in the country. It has been shown that individual invitation letters, especially with pre-booked appointment time and place, are very effective and result in high participation rates [10]. Very little is known about the cost-effectiveness of different invitation methods in Lithuania. Such information is very important for CC screening programme managers and healthcare providers and may help them to switch to more effective population-based cancer screening strategies using health resources efficiently.

This study aims to analyse the cost-effectiveness of personal invitation letter with appointment time and place, compared to the current practice of opportunistic primary healthcare provider initiative in Lithuania.

## 2. Materials and Methods

### 2.1. Data Collection and Study Design

The experimental randomized controlled trial was conducted in the PHCC of the Hospital of Lithuanian University of Health Sciences Kauno Klinikos in Lithuania. The study protocol was approved by the Lithuanian Bioethics Committee (protocol No BE-2-4 issued on 21/06/2017).

Before the study, the usual practice at PHCC was based on a verbal invitation by a family doctor or a nurse, who used to invite women to participate in CC screening programme during the appointments scheduled for other health issues. In 2014, the Department for Coordination of Preventive Programmes at the Hospital of Lithuanian University of Health Sciences Kauno klinikos was established. One of the functions of the Department is to introduce some elements of the organized CC screening programme on a pilot basis. The IT database was created by the Department together with the Kaunas University of Technology using the database of the National Health Insurance Fund and local hospital IT database to manage the invitation process. Personal invitation letter with pre-booked appointment time and place as well as information leaflet was prepared by the Department according to the European Guidelines for Quality Assurance in CC screening [5]. All eligible women registered at the PHCC and haven’t had screening for 3 or more years were invited personally by a letter to come to the PHCC to have a Pap smear test. If a woman still has not had a Pap smear test done within a year, a reminder letter was sent. Conventional Pap smear tests were evaluated at the Department of Pathology of the Hospital of Lithuanian University of Health Sciences Kauno klinikos. The results of Pap smear tests were classified according to the 2001 Bethesda system [12]. Abnormal Pap smear tests were subclassified into atypical squamous cells of undetermined significance (ASC-US), atypical squamous cells, cannot rule out high-grade lesion (ASC-H), low-grade squamous intraepithelial lesion (LSIL), high-grade squamous intraepithelial lesion (HSIL) and atypical glandular cell (AGS). After receiving written consent, women with a normal Pap smear test, inadequate cytological results (results that could not be determined) and unknown results (a Pap smear test was taken but no information about results) were informed by e-mail. Women with inadequate and unknown cytological results were invited to book an appointment for repeated Pap smear test. Women with abnormal Pap smear test results (ASC-US, ASC-H, LSIL, HSIL, AGS) were contacted by telephone by a nurse to book an appointment with specialist gynaecologist at the Hospital of Lithuanian University of Health Sciences Kauno Klinikos. 

The randomized controlled trial was launched on the 3rd of November 2014 (Figure 1). At that time overall 4357 women in the age group from 25 to 60 were registered at PHCC. All women with no registered routine Pap smear test during the previous 3 years were selected (n = 3294). These women were defined as non-attendees. The family doctors working at PHCC were randomly allocated to the experimental or control group together with non-attendees registered to the doctors. In the experimental group, all women (n = 1703) received a personal invitation letter by post inviting to participate in CC screening. A reminder letter was sent for a non-attendee if she did not respond to a screening invitation within a year. Both invitation letters included a pre-assigned appointment date, time and place to take a Pap smear test. Following a routine opportunistic screening practice in the control group, 817 women (out of 1703 eligible) were invited by family doctors to take a Pap smear test during women visits to the PHCC because of any other health issue. 

### 2.2. Invitation Strategies and Costs

A cost-effectiveness analysis was carried out based on a decision tree model structured in MS Office Excel 365 (Figure 2). The model was used to compare three invitation strategies: 

(a) The invitation strategy 1—a current opportunistic invitation strategy (regular practice); a verbal invitation by a family doctor to participate in the CC screening during the patients’ visits to the PHCC;

(b) The invitation strategy 2—a single personal postal invitation letter with scheduled appointment time and place to participate in the CC screening signed by the family doctor; 

(c) The invitation strategy 3—a personal postal invitation letter with scheduled appointment time and place with one reminder letter for non-attendees after one year. 

The invitation strategy 1 was compared with the invitation strategy 2 and then the strategy 2 was compared with the invitation strategy 3. The cost-effectiveness of each strategy was measured (1) per one additionally screened woman and (2) per one additional abnormal Pap smear test detected. The cost-effectiveness analysis has been carried out from the perspective of a healthcare provider. The direct healthcare costs included into the analysis were: programme costs reimbursed by the National Health Insurance Fund and overhead expenses of the hospital, related to the screening programme (such as the cost of salary of a family doctor, the screening department staff and a nurse related to the time consumed for screening procedures). 

### 2.3. The Cost of Healthcare Services

Description of different cost categories related to CC screening is shown in Table 1. The unit costs of invitation for preventive measure, taking a Pap smear test and laboratory assessment of cytological test were reimbursed by the National Health Insurance Fund. These costs were included in all invitation strategies. In the invitation strategy 1, the overhead expenses were related to the cost of time of a family doctor required for a verbal invitation of a woman to participate in the CC screening programme and for information of a woman about the result of Pap smear test during woman’s visit to a family doctor. In the invitation strategy 2 and 3, the overhead expenses were related to the cost of the time of the screening department staff for administrative purposes, preparation, and sending out the invitation letters as well as information about cytological test results (normal and inadequate) for a woman by e-mail. Besides, the cost of time for a PHCC nurse to inform a woman about abnormal results of Pap smear test by phone and to offer a registered appointment with a gynaecologist was taken into account. The cost of hourly pay for a family doctor, a screening department staff and a nurse was obtained from Lithuanian Official Statistics Portal and were used as their pay average. The cost of a personal invitation letter (i.e., a paper, printing, an envelope with a logo and postal fees) was not considered as additional expenses, because it was ascribed to the reimbursed cost of invitation services for preventive measures. All unit costs are presented in Euros referring to the 2015 prices. In invitation strategy 3, an annual discount rate of 5% was applied to expenses which are related to the services associated with a reminder letter.

### 2.4. Economic Analysis

The cost-effectiveness analysis was performed to estimate incremental cost-effectiveness ratios (ICERs) per one additionally screened woman and one additional abnormal cytological test (ASC-US, ASC-H, LSIL, HSIL) detected applying different invitation strategies. ICER represents a standard incremental analysis approach of difference in costs and difference in effects: $$ICER=\frac{Difference\;in\;expected\;costs\;between\;two\;invitation\;strategies}{Difference\;in\;expected\;result\;of\;the\;same\;invitation\;strategies\;}$$

ICER (1) shows the results in terms of Euros/per one additionally screened woman. The expected costs are the probability-weighted costs per one screened woman and the expected result is the probability of a woman being screened for each of the invitation strategies. ICER (2) shows the results in terms of Euros/ per one additional abnormal Pap smear test detected. The expected costs are the probability-weighted costs per one abnormal Pap smear test detected and the expected result is the probability of an abnormal Pap smear test for each of the invitation strategies.

A one-way sensitivity analysis was conducted to account for the uncertainty of costs including the variation which depends on discounting, salaries (family doctors, screening department staff and nurses) and invitation expenses. 

## 3. Results

### 3.1. The Participation in CC Screening and Pap Smear Test Results

Over the first year of the study, all women in the experimental group (n = 1591) were invited to participate in the screening by personal invitation letter. The participation rate in CC screening after the first invitation letter was 24.6% (Figure 2). After a reminder letter (n = 1042), an additional 16.9% of women attended for CC screening. Both invitation letters increased the coverage of CC screening in the experimental group to 35.6%.

In the control group, only 817 out of 1703 women were invited to participate in the CC screening by a family doctor. The participation rate was similar to that in the experimental group after the first invitation letter—25.8%. However, the coverage of CC screening in the control group was only 12.4% or twice lower than that in the experimental group. 

The frequency of abnormal cytology made up 25.8% after the first invitation letter, 22.2% after the reminder letter in the experimental group and 23.7% after a verbal invitation by a family doctor in the control group (Table 2). The proportions of different epithelial cell abnormalities were similar in all groups. 

### 3.2. The Cost-Effectiveness Analysis

In the first analysis of the base case scenario per one additionally screened woman, the invitation strategy 2 (a personal invitation letter) was more effective but more expensive than an opportunistic screening (the invitation strategy 1) (Table 3). ICER for the invitation strategy 2 versus strategy 1 made up €9.67 per one additionally screened woman. ICER for the invitation strategy 3 (a personal invitation letter with reminder letter), compared with the invitation strategy 2, was €13.47 per one additionally screened woman. In the second analysis of the base case scenario per one additional abnormal Pap smear test detected, ICER increased by €55.21 for the invitation strategy 2 versus current opportunistic invitation strategy (the strategy 1). ICER for the invitation strategy 3 increased by €86.88 per one additional abnormal Pap smear test detected if compared with the invitation strategy 2. Results of one-way sensitivity analysis are presented in Table 4. This analysis did not show a substantial difference from the cost-effectiveness results of the base case scenarios. 

## 4. Discussion

The study was conducted as a pilot project to evaluate a shift from an opportunistic to organised target population invitation model to improve the coverage and to optimise the cost of CC screening. The cost-effectiveness of alternative strategies: a verbal invitation by a family doctor, a personal invitation letter and an invitation letter with a reminder letter for non-attendees, has been evaluated at PHCC of the Hospital of Lithuanian University of Health Sciences Kauno Klinikos in Lithuania. The study, carried out from a healthcare provider’s perspective, has shown that the systematic invitation letter with scheduled appointment time and place approach resulted in about twice higher proportion of invited and attended women, as well as two-fold more abnormal Pap smear tests, were detected. However, it incurred additional expenses (€9.67 per one additionally screened woman and €55.21 per one additional abnormal Pap smear test detected) in comparison with a regular practice based on opportunistic invitation. A systematic personal invitation letter with a reminder letter for non-attendees additionally increased the coverage of CC screening and the proportion of abnormal Pap smear tests detected, but it was more expensive than sending a single personal invitation letter. ICER increased by 1.4 times per one additionally screened woman and by 1.6 times per one additional abnormal Pap smear test detected. 

An increase in the coverage of CC screening should be a prime purpose of the healthcare system [13]. Scientific evidence suggests that well-organized population-based CC screening programmes using personal invitation letters approach ensure greater participation of the target population than the opportunistic invitation model [7,10]. Adequate coverage and participation rates are very important parameters of programme performance and effectiveness. Evidence from EU countries, where organized screening programmes for CC was implemented earlier, demonstrates their impact on reducing the incidence of and mortality from the disease [6]. According to the EU Council recommendations, high quality of well-organized CC screening should be assured through the identification and a personal invitation (call/recall system) of each woman of the target population, the performance of a screening test, a follow-up, the treatment of detected abnormalities, monitoring and evaluation of the effectiveness of the programme over time [5]. In opportunistic screening, the participation strongly depends on the frequency of visits to a family doctor and doctors’ active involvement in the provision of information about screening [14]. In Lithuania, visits of a woman to a PHCC are free of charge if she is a patient of the clinic. Because of the prevailing cultural and behavioural traditions as well as easy access to specialist care women tend to have their gynaecologist, who can take Pap smear test free of charge. Favourable reimbursement conditions of cytological test performed outside of the programme predetermine the high prevalence of this opportunistic screening practice. Besides, Lithuania is the only country in the EU with a population-based screening programme, but no screening registry [14,15]. The data of invitations, Pap smear tests, their results and visits to family doctors for information about CC screening results are registered on the patient database of the PHCC which is synchronized with the database of the National Health Insurance Fund because of reimbursement, but not clinical monitoring purposes. The lack of an organized invitation system, inappropriate monitoring of screening programme performance, and the absence of registration of opportunistic screening outside the programme are important drawbacks of the programme in Lithuania [14,15]. After the introduction of the national CC screening programme in Lithuania, some intermediate goals have been achieved [16]. The rates of detection of premalignant lesions (carcinoma in situ) and the number of cases detected at CC early stages have increased, while the incidence of stage III and IV CC has decreased. 

Cervical cancer is a highly preventable disease with the help of a cytological screening programme that detects and treats precancerous lesions and, as a result, reduces the number of cancer cases that require treatment. Prevention through vaccination and screening of cancer costs less in comparison with the treatment of cancer [17]. The improvement of the effectiveness of CC screening may require additional expenses [18]. CC screening implies multiple trade-off and decision-makers have to decide whether they are willing to pay and accept additional expenses to optimise the strategy. Sustained effort in CC screening finally results in the drop of expenses in the other fields of health services [19]. Cost-effectiveness of screening depends on how many additional pre-cancer and early stages of cancer cases are detected. In this study, cost-effectiveness has been measured per screened woman and per an abnormal Pap smear test found in a women population served by a single PHCC. Advantage of a systematic invitation still has to be assessed in a broader perspective to determine its effect on a national scale. 

To our knowledge, similar studies are lacking. It is complicated to compare the cost of different invitation strategies across the EU countries with different population-based screening policies, screening tests (conventionally cytology, liquid-based cytology, human papillomavirus DNA test), and healthcare systems [20]. The application of various invitation strategies can increase participation in the preventive programme. The choice of the most cost-effective strategy is important for reaching the best possible effect using resources efficiently. Not all invitation strategies are equally effective. The differences in the components of the strategies can result in large differences in their cost-effectiveness that depends on the attendance rate, the cost of a Pap smear test, the prevalence of abnormal test, etc. A study in Spain, where cytological screening for CC is largely opportunistic, has shown that an invitation letter is the most cost-effective intervention if compared with opportunistic screening [21]. The ICER made up €2.78 per 1% increase in the screening coverage. Participation after an invitation letter with a phone reminder was higher than after an invitation letter—23% and 18.6% respectively, but more expensive if compared with opportunistic screening. Heranney et al. have found that a reminder letter after a previous invitation letter to a CC screening was as effective as a telephone call and less expensive [22]. In contrast, Broberg et al. have demonstrated that an invitation by phone to remind long-term non-attendees significantly increases participation in a well-running screening programme, leads to a noticeable increase in the detection of abnormal Pap smear tests and does not seem to increase costs, in comparison with a reminder letter [23]. The recent studies have assessed the effect of vaginal self-sampling test for high-risk human papillomavirus (HPV) types at home as an alternative to reminder letter to increase participation among non-attendees in organized CC screening [24,25,26]. It was shown, that HPV testing started at age 30 years or later and repeated at 5-year intervals, increases participation among non-attendees and seems to be the most cost-effective strategy of CC screening [25]. Lithuania is in the process of introducing HPV-based screening for women older than 35 years. The effective invitation strategy should be helpful in increasing awareness and motivation of women to participate in CC screening.

This is the first study in Lithuania that was carried out as the cost-effectiveness analysis of different invitation strategies in the CC screening programme. The study has several limitations. First, this cost-effectiveness analysis was based on data of a single PHCC. Second, the analysis included only direct ambulatory healthcare expenditure. Third, the intermediate outcomes (the proportion of screened women and abnormal cytological results detected) were used for calculations. Besides, some costs related to systematic invitation might be lower if such invitation strategy were implemented on a regional or national level in Lithuania. 

## 5. Conclusions

A personal invitation letter with scheduled appointment time and place approach is more effective in increasing participation rate in CC screening and the number of detected abnormal Pap smear tests; however, it incurs additional cost compared with an opportunistic invitation model. A systematic personal invitation letter with an additional reminder letter for non-attendees additionally has increased the coverage of the total CC screening and the detection of abnormal Pap smear tests, but it was much more expensive than a single personal invitation letter. Further studies for a longer period and larger population are needed to decide on the most cost-effective approach of the systematic invitation strategy in CC screening on a national level in Lithuania. 

## Figures and Tables

**Figure 1 ijerph-16-05035-f001:**
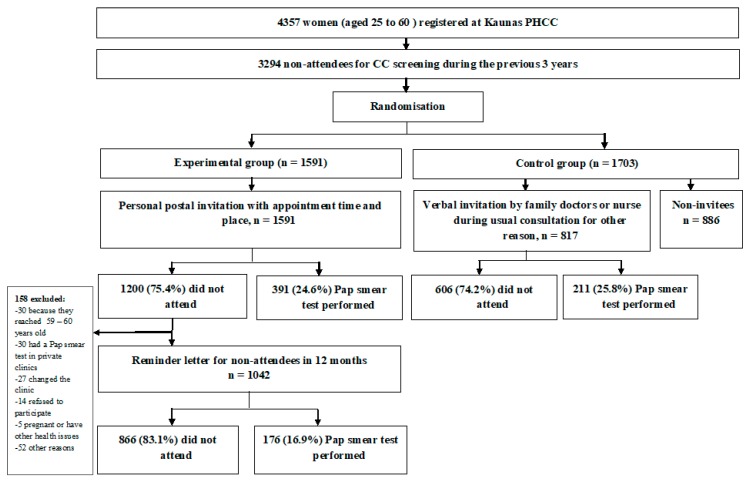
The scheme of the study.

**Figure 2 ijerph-16-05035-f002:**
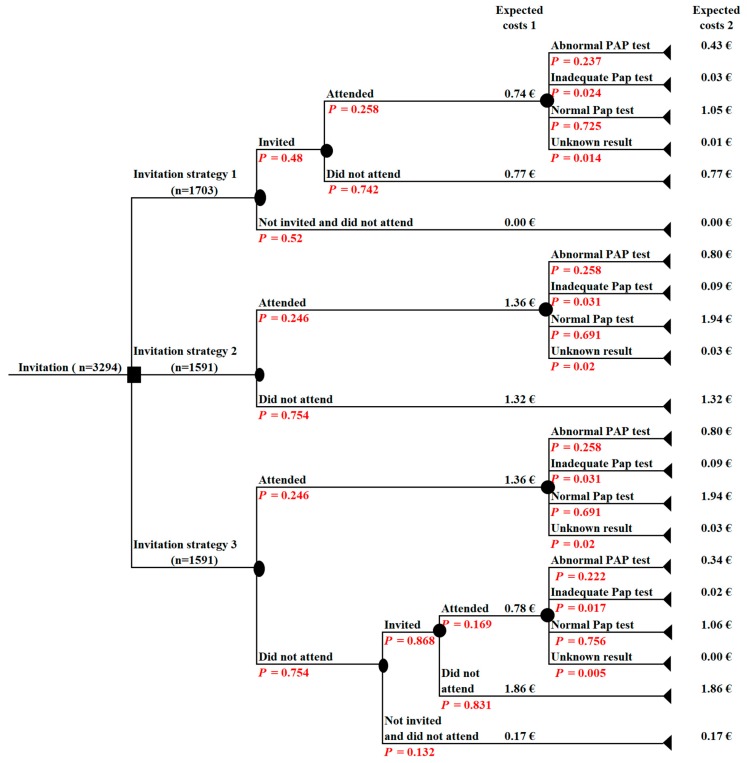
A decision tree for cost-effectiveness analysis of different screening strategies among non-attendees (base case scenario). Expected cost 1—the probability-weighted cost for invitation plus cost for Pap smear taking per one woman; expected cost 2—expected cost 1 plus the probability-weighted cost for laboratory analysis and cost of providing information about test results per one woman.

**Table 1 ijerph-16-05035-t001:** Description of different cost categories.

Cost Category	Description	Unit Cost (€)
A verbal invitation from a family doctor	Reimbursed cost of the invitation	1.45
	Cost of time spent for a verbal invitation by a family doctor (calculated using an average of hourly wage rate).	0.70
A personal invitation letter	Reimbursed cost of the invitation	1.45
	Cost of time of the screening department staff for identification of unscreened women, preparation and sending out of invitation letters (calculated using hourly wage rate).	0.30
A personal reminder letter	Cost of a personal invitation letter discounted by 5%.	1.67
Pap smear test-taking	Reimbursed cost	3.79
Conventional Pap smear test assessment	Reimbursed cost	5.79
Information about the Pap smear test result	Cost of time spent (calculated using hourly wage rate):	
	a family doctor	2.80
	screening department staff	0.08
	a nurse	1.28

**Table 2 ijerph-16-05035-t002:** Pap smear test results by intervention group.

Pap Smear Test Results	Experimental Group		Control Group
	The First Invitation Letter (n = 391)	The Reminder Letter (n = 176)	Invitation by a Family Doctor (n = 211)
	n (%)	n (%)	n (%)
Unknown ^a^	8 (2.0)	1 (0.6)	3 (1.4)
Inadequate ^b^	12 (3.1)	3 (1.7)	5 (2.4)
Normal	270 (69.1)	133 (75.6)	153 (72.5)
Abnormal ^c^	101 (25.8)	39 (22.2)	50 (23.7)
ASC-US	42 (10.7)	14 (7.9)	17 (8.1)
ASC-H	-	1 (0.6)	1 (0.5)
LSIL	59 (15.1)	24 (13.6)	30 (14.2)
HSIL	-	-	2 (0.9)

^a^ The Pap smear has been taken but the cytologic test results are unknown; ^b^ result that could not be determined; ^c^ epithelial cell abnormalities: atypical squamous cells of undetermined significance (ASC-US), atypical squamous cells, cannot rule out high-grade lesion (ASC-H), low grade squamous intraepithelial lesion (LSIL), high grade squamous intraepithelial lesion (HSIL).

**Table 3 ijerph-16-05035-t003:** The direct cost of services associated with cervical cancer (CC) screening.

Cost Category	Unit Cost (€)	Invitation Strategy 1 (n = 1703)		Invitation Strategy 2 (n = 1591)		Invitation Strategy 3 (n = 1591)	
		n	Cost (€)	n	Cost (€)	n	Cost (€)
A verbal invitation from a family doctor	2.15	817	1756.55				
A personal invitation letter	1.75			1591	2784.25	1591	2784.25
A personal reminder letter	1.67					1042	1740.14
Pap smear test-taking	3.79	211	799.69	391	1481.89	567	2117.25
Conventional Pap smear test assessment	5.79	204	1181.16	383	2217.57	558	3181.82
Information about the Pap smear test result:							
family doctor	2.80	70	196				
screening department staff	0.08			290	23.2	427	34.16
nurse	1.28			101	129.28	140	176.86
Cost of screened women: invitation + performed Pap smear test			2556.24		4266.14		6641.64
Total cost			3933.40		6636.19		10034.48
Expected cost per one screened woman			1.51		2.68		4.17
Expected cost per one abnormal Pap smear test detected			2.29		4.18		6.31
Probability of a woman being screened *		0.124		0.246		0.357	
Probability of abnormal Pap smear test detected *		0.029		0.063		0.088	
ICER per one additionally screened woman					9.67		13.47
ICER per one additional abnormal Pap smear test detected					55.21		86.88

* A probability of a particular branch was estimated by multiplying all of the probabilities related with the particular pathway (Figure 2). The invitation strategy 1—invitation by a family doctor; the invitation strategy 2—invitation letter; invitation strategy 3—invitation letter plus reminder letter; ICER - incremental cost-effectiveness ratio.

**Table 4 ijerph-16-05035-t004:** Results of sensitivity analysis.

Scenario	Invitation Strategy	ICER (1)	ICER (2)
Base case	Strategy 2 (invitation letter) versus strategy 1 (regular practice)	€9.67	€55.21
	Strategy 3 (invitation letter + reminder letter) versus strategy 2 (invitation letter)	€13.47	€86.88
Without the discounting of a reminder letter	Strategy 2 (invitation letter) versus strategy 1 (regular practice)	€9.67	€55.21
	Strategy 3 (invitation letter + reminder letter) versus strategy 2 (invitation letter)	€14.15	€91.23
Without salary expenses	Strategy 2 (invitation letter) versus strategy 1 (regular practice)	€9.96	€55.86
	Strategy 3 (invitation letter + reminder letter) versus strategy 2 (invitation letter)	€11.78	€77.78
Expenses of invitation services decrease by 20%	Strategy 2 (invitation letter) versus strategy 1 (regular practice)	€8.43	€50.79
	Strategy 3 (invitation letter + reminder letter) versus strategy 2 (invitation letter)	€11.89	€79.62
Expenses of invitation services increase by 20%	Strategy 2 (invitation letter) versus strategy 1 (regular practice)	€10.90	€59.63
	Strategy 3 (invitation letter + reminder letter) versus strategy 2 (invitation letter)	€15.12	€94.33

Abbreviations: ICER (1)—incremental cost-effectiveness ratio per one additionally screened woman; ICER (2)—incremental cost-effectiveness ratio per one additional abnormal Pap smear test detected.

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
