# Peer review of "The Cost-Effectiveness Analysis of Cervical Cancer Screening Using a Systematic Invitation System in Lithuania"

_ijerph, 2019, doi:10.3390/ijerph16245035_

Round 1

Reviewer 1 Report

The paper by Paulauskiene et al reports the results of a cost-effectiveness analysis comparing three different invitation strategies for cervical cancer screening; 1) the invitation by family doctor at the time of a visit (modality in use), 2) the personal invitation by letter, and 3) the personal invitation by letter with a reminder letter for non attenders. By evaluating the incremental cost-effectiveness ratio (ICER) per one additionally screened woman and per one additional abnormal Pap smear test detected, they conclude that a higher participation rate is achieved by personal invitation letters, but at higher expenses compared to the modality in use.

This conclusion confirms the data from other countries, but was not previously analyzed for Lithuania.

Since the analysis assessed the cost-effectiveness from the perspective of a healthcare provider, additional data and considerations could be useful to strenghten the benefits of an organized invitation strategy.

The additional costs of strategy 2 in comparison to strategy 1 derive from the higher number of women invited and tested (the % of attenders was not different), that translate into higher coverage and reduction/avoidance of inequalities. To assess whether the two strategies reached women with a different attitude towards screening practices (i.e., whether different proportions of non attenders were represented in the two groups), it should be useful to compare the cytological diagnoses of the abnormal Pap smears in the two groups.

The same considerations apply for the additional costs of strategy 3 in comparison to strategy 2 since the remind letter obtains the real gain in terms of attending women; comparison of the cytological diagnoses of the abnormal Pap smears could give some insight into the characteristics of the respondent women.

I do agree that the same cost-effectiveness analysis should be perfomed on a larger scale, and that a quality monitoring system of the screening process should be implemented.

Author Response

Response to Reviewer 1 Comments

Point 1: The additional costs of strategy 2 in comparison to strategy 1 derive from the higher number of women invited and tested (the % of attenders was not different), that translate into higher coverage and reduction/avoidance of inequalities. To assess whether the two strategies reached women with a different attitude towards screening practices (i.e., whether different proportions of non attenders were represented in the two groups), it should be useful to compare the cytological diagnoses of the abnormal Pap smears in the two groups.

The same considerations apply for the additional costs of strategy 3 in comparison to strategy 2 since the remind letter obtains the real gain in terms of attending women; comparison of the cytological diagnoses of the abnormal Pap smears could give some insight into the characteristics of the respondent women.

Response 1: Thank you for favourable evaluation of our manuscript and helpful comments. Following the reviewer’s advice, we rearrange the Table 2 and included the distribution of the abnormal Pap smears by different cell abnormalities (page 6).

Also, we added a sentence in the Results section in the 3rd paragraph: ‘The proportions of different epithelial cell abnormalities were similar in all groups.’

Reviewer 2 Report

In this study, the authors conducted an economic analysis comparing several invitation strategies they designed to improve the participation rate in cervical cancer screening. Overall, the introduction and discussion were well developed, but methods should be better stated, and several issues about the comparableness between the participant groups need to be clarified. Please see below for my comments in detail.

Line 36, change “The 2nd December 2003” to “In 2003,”. Line 65, change “randomized control trial” to “randomized controlled trial”. Overall, study design was stated clearly, but improvement may be made for the statement of statistical analysis. In specific: Line 143, to make this paper more readable for the general audience, it should be better to give a little bit more detailed description of the definitions and methods used, e.g. how the incremental cost-effectiveness ratios (ICERs) were calculated, what indications that ICERs can provide. Also, the categories of the abnormal cytological test should be described in a bit more detail. Another of my concerns is the comparableness of the participants in the two sets of comparisons: strategy 2 vs 1; strategy 3 vs 2. First, for the control group, why only less 50% (817/1703) of the patients were verbally invited, maybe these invited patients had particular characteristics compared with those not invited and the experimental group. This may result in an incomparableness between the control group (1703) and the experimental group (1591). Second, for the “experimental group”, those taking strategy 2 and 3 may be not comparable either. Because the participants for the strategy 3 may be more likely to refuse to participate the screening, again, this may result from some particular characteristics, e.g. they are healthier, or younger than those accepting the first invitation, who were counted in those taking strategy 2. The authors should clarify these issues.

Author Response

Response to Reviewer 2 Comments

Thank you for your helpful comments for our manuscript. We hope that we have successfully addressed all of the concerns raised, and we believe that the manuscript has been substantially improved. Our detailed responses to the comments and the description of the changes we have made to the manuscript are provided below.

Point 1: Line 36, change “The 2nd December 2003” to “In 2003,”

Response 1: Corrected

Point 2: Line 65, change “randomized control trial” to “randomized controlled trial”.

Response 2: Corrected

Point 3: Line 143, to make this paper more readable for the general audience, it should be better to give a little bit more detailed description of the definitions and methods used, e.g. how the incremental cost-effectiveness ratios (ICERs) were calculated, what indications that ICERs can provide.

Response 3: To explain the calculations in cost-effectiveness analysis we added to the Methods section Economic analysis subsection:

ICER represents a standard incremental analysis approach of difference in costs and difference in effects:

ICER =  (Difference in expected costs between two invitation strategies) :        (Difference in expected results of the same invitation strategies)

ICER (1) shows the results in terms of Euros/per one additionally screened woman. The expected costs are the probability-weighted costs per one screened woman and the expected result is the probability of a woman being screened for each of the invitation strategies. ICER (2) shows the results in terms of Euros/ per one additional abnormal Pap smear test detected. The expected costs are the probability-weighted costs per one abnormal Pap smear test detected and the expected result is the probability of an abnormal Pap smear test for each of the invitation strategies.

Furthermore, some explanations were added to the legend of Figure 2 and some corrections were made in Table 3.

Point 4: The categories of the abnormal cytological test should be described in a bit more detail.

Response 4: Following the reviewer’s advice, we described the categories of the abnormal cytological test classified according to the 2001 Bethesda system in the Methods section: ‘Abnormal Pap smear tests were categorized as atypical squamous cells of undetermined significance (ASC-US); atypical squamous cells, cannot rule out high-grade lesion (ASC-H); low grade squamous intraepithelial lesion (LSIL); high grade squamous intraepithelial lesion (HSIL) and atypical glandular cell (AGS).

 Point 5: Why only less 50% (817/1703) of the patients were verbally invited, maybe these invited patients had particular characteristics compared with those not invited and the experimental group. This may result in an incomparableness between the control group (1703) and the experimental group (1591).

Response 5: We would like to explain that the proportion of women invited in the control group reflects the existing practice in Lithuania to invite women to participate in the screening programme only when she attends the primary health centre due to any other health issues. As long as younger women (25-50 yrs.) tend to have few health problems, they do not visit doctors and therefore do not receive invitations to participate in the screening. So the proportion of invited women in the control group was much lower compared to the experimental group where all women received the invitation letters. In the cost-effectiveness analysis, the lower screening coverage in the control group has been taken into account.

We made some changes in the Methods section to clarify the differences in the invitation strategies between the experimental and control group: ‘The family doctors working at PHCC were randomly allocated to the experimental or control group together with non-attendees registered to the doctors. In the experimental group, all women (n=1703) received a personal invitation letter by post inviting to participate in CC screening. A reminder letter was sent for a non-attendee if she did not respond to a screening invitation within a year. Both invitation letters included a pre-assigned appointment date, time and place to take a Pap smear test. Following a routine opportunistic screening practice in the control group, 817 women (out of 1703 eligible) were invited by family doctors to take a Pap smear test during women visits to the PHCC due to any other health issue.’

No differences in age structure between invited women in the experimental and control group were found. Furthermore, the participation rate and the proportion of abnormal Pap smears were very similar in both groups. So, we assume that the groups are comparable.

Point 6: Second, for the “experimental group”, those taking strategy 2 and 3 may be not comparable either. Because the participants for the strategy 3 may be more likely to refuse to participate the screening, again, this may result from some particular characteristics, e.g. they are healthier, or younger than those accepting the first invitation, who were counted in those taking strategy 2.

Response 6: We agree with the reviewer that women targeted by the strategy 3 might differ from women covered by the strategy 2 because they are less motivated to participate in preventive activities (they did not participate in the screening before the experiment and did not respond to the first invitation letter). However, this is a real situation. Comparing the strategy 2 and strategy 3, we tested the cost-effectiveness of the reminder letter; that is whether or not to send the second invitation letter for those women who have little interest in participating. We found that some of those women responded after the reminder letter; however, costs were much higher compared to one invitation letter.

Reviewer 3 Report

Cervical cancer (CC) can be prevented by having regular screenings to find any precancers and treat them. The success of screening programme depends on screening coverage. Screening strategies differ between countries. Some countries have organized screening that systematically tests all women in the defined target group, either on a national or regional level.

Paulauskiene et al have used a decision tree to compare an opportunistic invitation by a family doctor, a personal postal invitation letter with appointment time and place, and a personal postal invitation letter with appointment time and place with one reminder letter. Cost-effectiveness was defined as an incremental cost-effectiveness ratio (ICER) per one additionally screened woman and per one additional abnormal Pap smear test detected. The ICER of one personal postal invitation letter was €9.67 per one additionally screened woman and €55.21 per one additional abnormal Pap smear test detected in comparison with the current screening practice. The ICER of a personal invitation letter with an additional reminder letter compared to one invitation letter was €13.47 and €86.88 respectively. In conclusion, a personal invitation letter approach is more effective in increasing the participation rate in CC screening and the number of detected abnormal Pap smears; however, it incurs additional expenses compared with current invitation practice.

The claims are properly placed in the context of the previous literature. The experimental data support the claims. The manuscript is written clearly enough that most of it is understandable to non-specialists. The authors have provided adequate proof for their claims, without overselling them. The authors have treated the previous literature fairly. The paper offers enough details of methodology so that the experiments could be reproduced.

Comments

More and more countries are now switching from cytology based screening to primary HPV screening. I am not sure the best strategy for Lithuania is to start organized cytology screening with personal postal invitation letter with appointment time and place. I think it would be better send postal equipment for vaginal self-sampling test for HPV. Only women with a positive HPV-test need an appointment time and place. In the Netherlands it is recommended that women with negative HPV-tests can be screened at age 30, 35, 40, 50 and 60 (five screenings through lifetime). It is possible to increase screening coverage more using self sampling than postal invitation letter. HPV-testing is more sensitive than cervical cytology and may prevent more cases of cervical cancers with fewer screening rounds during lifetime and less costs.

Author Response

Response to Reviewer 3 Comments

Thank you for favourable and encouraging evaluation of our manuscript. We absolutely agree with your comment regarding the benefits of introducing HPV based screening. Right now, our country is in the process of introducing HPV screening for women older than 35 yrs. However, we think that effective invitation strategy should be helpful in increasing awareness and motivation of women to participate. The Pilot study of HPV screening showed that women, especially younger, were satisfied with this screening method; however, the participation rate was still unsatisfactory.

We added to the Discussion section several sentences about HPV screening: ‘The recent studies have assessed the effect of vaginal self-sampling test for high-risk human papillomavirus (HPV) types at home as an alternative to reminder letter to increase participation among non-attendees in organized CC screening [24-26]. It was shown, that HPV testing started at age 30 years or later and repeated at 5-year intervals, increases participation among non-attendees and seems to be the most cost-effective strategy of CC screening [25]. Lithuania is in the process of introducing HPV based screening for women older than 35 years. The effective invitation strategy should be helpful in increasing awareness and motivation of women to participate in CC screening.’